# In Vivo Microdialysis of Endogenous and ^13^C-labeled TCA Metabolites in Rat Brain: Reversible and Persistent Effects of Mitochondrial Inhibition and Transient Cerebral Ischemia

**DOI:** 10.3390/metabo9100204

**Published:** 2019-09-27

**Authors:** Jesper F. Havelund, Kevin H. Nygaard, Troels H. Nielsen, Carl-Henrik Nordström, Frantz R. Poulsen, Nils. J. Færgeman, Axel Forsse, Jan Bert Gramsbergen

**Affiliations:** 1VILLUM Center for Bioanalytical Sciences, Department of Biochemistry and Molecular Biology, University of Southern Denmark, Campusvej 55, 5230 Odense M, Denmark; nils.f@bmb.sdu.dk; 2Department of Neurosurgery, Odense University Hospital, University of Southern Denmark, Sdr. Boulevard 29, 5000 Odense C, Denmark; kevin.nygaard@hotmail.com (K.H.N.); troels.nielsen@rsyd.dk (T.H.N.); carl-henrik.nordstrom@med.lu.se (C.-H.N.); frantz.r.poulsen@rsyd.dk (F.R.P.); axel.forsse@gmail.com (A.F.); 3BRIDGE—Brain ResearchE—Inter-Disciplinary Guided Excellence, Institute of Clinical Research, University of Southern Denmark, Winsløwparken 19, 5000 Odense C, Denmark; 4Institute of Molecular Medicine, University of Southern Denmark, 55, 5230 Odense C, Denmark; jbgramsbergen@health.sdu.dk

**Keywords:** ^13^C-labeled succinate, cerebral ischemia, energy metabolism, endothelin-1, LC-MS, malonate, micro-dialysis, mitochondrial dysfunction, reperfusion, tricarboxylic acid cycle

## Abstract

Cerebral micro-dialysis allows continuous sampling of extracellular metabolites, including glucose, lactate and pyruvate. Transient ischemic events cause a rapid drop in glucose and a rise in lactate levels. Following such events, the lactate/pyruvate (L/P) ratio may remain elevated for a prolonged period of time. In neurointensive care clinics, this ratio is considered a metabolic marker of ischemia and/or mitochondrial dysfunction. Here we propose a novel, sensitive microdialysis liquid chromatography-mass spectrometry (LC-MS) approach to monitor mitochondrial dysfunction in living brain using perfusion with ^13^C-labeled succinate and analysis of ^13^C-labeled tricarboxylic acid cycle (TCA) intermediates. This approach was evaluated in rat brain using malonate-perfusion (10–50 mM) and endothelin-1 (ET-1)-induced transient cerebral ischemia. In the malonate model, the expected changes upon inhibition of succinate dehydrogenase (SDH) were observed, i.e., an increase in endogenous succinate and decreases in fumaric acid and malic acid. The inhibition was further elaborated by incorporation of ^13^C into specific TCA intermediates from ^13^C-labeled succinate. In the ET-1 model, increases in non-labeled TCA metabolites (reflecting release of intracellular compounds) and decreases in ^13^C-labeled TCA metabolites (reflecting inhibition of de novo synthesis) were observed. The analysis of ^13^C incorporation provides further layers of information to identify metabolic disturbances in experimental models and neuro-intensive care patients.

## 1. Introduction

In neuro-intensive care units, cerebral microdialysis is routinely used to monitor interstitial levels of glucose, lactate and pyruvate, the main energy substrates of the brain. In patients with traumatic brain injury (TBI) or aneurysmal subarachnoid hemorrhage (aSAH), periods of compromised cerebral blood flow are characterized by a decrease in glucose, a rise in lactate and elevated L/P ratios [1,2]. Elevated L/P ratios with normal or increased levels of pyruvate after an ischemic insult are considered a metabolic marker of mitochondrial dysfunction, which has been associated with delayed neurological deterioration (DND) [3,4]. Early detection and monitoring of mitochondrial dysfunction, as well as understanding the underlying mechanisms of disturbed energy metabolism post-injury are essential to improve treatments and outcome in neuro-intensive care patients.

Brain energy metabolism has been studied in vitro and in vivo by perfusion with ^14^C-labeled (radioactive) energy substrates, including glucose, lactate, pyruvate, glutamate and glutamine, and monitoring ^14^CO_2_ production under various experimental conditions. Studies using fluorocitrate, which at lower concentrations inhibits glial, but not neuronal TCA cycle activity, combined with ^14^C-labeled microdialysis allowed assessment of oxidation rates of different energy substrates in glial cells or neurons [5].

More recently cerebral microdialysis and perfusion with ^13^C-labeled substrates and analysis of ^13^C-labeled TCA intermediates has been used to study brain energy metabolism in head injury patients [6,7,8]. ^13^C-labeled energy substrates included 1,2-^13^C_2_ glucose to study glycolysis and the pentose phosphate pathway [6], 3-^13^C-lactate to study lactate metabolism via the TCA cycle [8] and 2,3-^13^C_2_-succinate to study enhancement of TCA cycle metabolism [7]. In the clinical microdialysis studies mentioned above, the recovered ^13^C-labeled metabolites were analyzed by nuclear magnetic resonance (NMR). Although NMR is a standard technique for identifying ^13^C-labeled molecules, the sensitivity is not very high, meaning that large volumes of dialysate are needed and only abundant metabolites in the millimolar range can be detected. In the study by Jalloh et al. 2016, perfusing with 12 mM 2,3-^13^C_2_-succinate (a pharmacologically active dose enhancing local brain metabolism), micro-dialysate samples were pooled over a 24 h period (180 µL pooled dialysates) to allow detection of ^13^C-labeled fumarate, malate, lactate and glutamine. As that study showed, exogenous succinate was taken up by brain cells (astrocytes and neurons) and metabolized via the TCA cycle within mitochondria.

In the present experimental study in rats, we used a similar approach of perfusing brain tissue with ^13^C-labeled succinate through the dialysis probe (Figure 1) and measuring ^13^C-labeled TCA-centered metabolites in the dialysate. However, in contrast to the study by Jalloh et al., we used highly sensitive LC-MS to identify and quantify ^13^C-labeled TCA-centered metabolites upon continuous perfusion with a tracer dose (1 mM) of uniformly ^13^C-labeled succinate. Because of the higher sensitivity of LC-MS, we were able to measure both endogenous (^12^C) and ^13^C-labeled TCA-centered metabolites in striatal dialysates with a temporal resolution as low as 30 min (30 µL samples). We only show TCA-centered metabolites, however, many other endogenous compounds, e.g., amino acids, purines and pyrimidines were detected. The major improvement in time resolution and extra layers of information (acute release of endogenous metabolites and acute changes of de novo synthesis) that this approach offers, allows detailed biochemical monitoring and better understanding of mechanisms causing mitochondrial dysfunction and DND in TBI and aSAH patients. Finally, we want to emphasize that the method described here can be adapted to monitor metabolic disturbances following physical or medical interventions in other tissues, for instance subcutaneously or intramuscularly, and under various pathological conditions, such as diabetes or cancer.

In this study in living brain the ^13^C-labeled microdialysis LC-MS approach was validated using two different rat models:(a)Mitochondrial dysfunction induced by local perfusion with malonate, a reversible inhibitor of SDH.(b)Transient cerebral ischemia induced by intracerebral application of the potent vasoconstrictor ET-1.

## 2. Results

### 2.1. The malonate Model

Perfusion with the SDH inhibitor malonate (10 and 50 mM at 0 h and 15 h, respectively) caused a very clear dose-dependent increase in endogenous succinate (Figure 2). Other TCA metabolites showed the opposite effect: A dose-dependent decrease in abundance. Changes in glutamine were related to those in alpha-ketoglutarate. Changes in ^13^C-incorporated metabolites showed tendencies similar to endogenous metabolites, but the effects were generally much more pronounced (Figure 2).

### 2.2. Transient Cerebral Ischemia Model

In the ET-1-induced cerebral ischemia-reperfusion model we observed a glucose drop below 50% of baseline levels in the first 30 min fraction. Large increases of glucose-6-phosphate (up to 800%) and lactate (up to 300%) were observed within in the first 2 h after induction of ischemia and did not normalize completely in the subsequent hours of monitoring (Figure 3).

We observed large differences in the magnitude and timing of change in endogenous and ^13^C-labeled metabolites following ischemia-reperfusion (Figure 3). Changes in the abundance of ^13^C-fumarate and ^13^C-malate show the opposite tendency compared to their ^12^C analogues, which illustrates that the subcellular source and de novo synthesis of ^12^C or ^13^C-labeled compounds differ, i.e., ^12^C can be derived from both cytosol or mitochondrial compartments whereas ^13^C is only derived from mitochondria. Further down-stream citrate/isocitrate showed increases in both endogenous and ^13^C-labeled compounds with the latter showing the most dramatic changes. There were no apparent differences in alterations between endogenous and ^13^C-labeled alpha-ketoglutarate (increases up to 200%) and the related compound glutamine (no changes after the insult).

Alterations in pyruvate, which can be derived from glycolysis or formed by decarboxylation of malate, differed according to labeling pattern, i.e., an increase in ^13^C-pyruvate was observed after ischemia-reperfusion whereas the level of the ^12^C form was not altered (Figure 3).

### 2.3. ^13^C-labeling %

In Table 1, ^13^C-labeling %, defined as ^13^C-labeled/total compound (^13^C + ^12^C) × 100% for several TCA intermediates and the monocarboxylates pyruvate and lactate are shown during baseline conditions, the ischemic period and 30 min reperfusion, as well as after longer reperfusion time following ET-1-induced vasospasm. Fumarate and malate show the highest labeling % whereas isocitrate, alpha-ketoglutarate and glutamine show a considerably lower labeling %. The labeling pattern of TCA metabolites is in agreement with the biochemical distance to the labeling source, i.e., ^13^C-succinate.

After ischemia and 30 min reperfusion there is a significant drop in labeling % for fumarate and malate, indicating mitochondrial dysfunction, which has largely recovered after > 4 h reperfusion. Interestingly, labeling % for pyruvate increased after ischemia, but did not reach statistical significance.

### 2.4. Histological Brain Damage

In Figure 4 placement of the microdialysis probes in the malonate (A) and ET-1 model (B) is shown, as well as the guide cannula for ET-1 infusion (B). Malonate perfusion did not cause histological brain damage whereas ET-1 infusion caused ischemic damage in the ipsilateral striatum.

## 3. Discussion

### 3.1. General

In this study we showed the potential of studying acute mitochondrial dysfunction in living brain by perfusion with a tracer dose of ^13^C-labeled succinate through a microdialysis probe and subsequent LC-MS analysis of TCA-centered metabolites in the dialysates. Since LC-MS allowed the detection and relative quantification of both ^13^C-labeled and endogenous (^12^C) TCA metabolites, we can distinguish the efflux of endogenous metabolites as a consequence of cellular damage from changes in *de novo* synthesis (^13^C-labeled metabolites) as a result of mitochondrial inhibition.

We assume that the efflux of ^13^C-succinate from the probe and delivery to the cells is similar under baseline and experimental conditions, because perfusion with malonate or induction of cerebral ischemia did not cause any changes in ^13^C-succinate levels in the dialysates. In contrast, ^13^C-labeling of other TCA metabolites was strongly influenced by the experimental conditions. Succinate uptake into glial cells and neurons is mediated by the SLC13 family of Na^+^-coupled dicarboxylate and tricarboxylate transporters [9]. Such transporters are found both on cell membranes and the mitochondrial inner membrane. Succinate is metabolized to fumarate by SDH localized on the inner mitochondrial membrane and is also known as electron transport chain complex II. Thus, all ^13^C-labeled metabolites found in the dialysate are the result of succinate metabolism in the TCA cycle.

Perfusion with ^13^C_4_-labeled succinate resulted in a labeling % of ^13^C-labeled versus endogenous metabolites in accordance with the direction of the TCA cycle and “biochemical distance” (relationship) to succinate (see Figure 3 for full labeling patterns after one round of the TCA cycle and Table 1 for labeling efficacy, % full labeling). Thus, the highest labeling % was found for the TCA metabolites fumarate and malate, followed by citrate/isocitrate and alpha-ketogluarate. ^13^C-alpha-ketogluarate can be converted to ^13^C-glutamate (not detected), which again can be converted to ^13^C-glutamine by glutamine synthase in astrocytes. ^13^C-pyruvate can be formed by decarboxylation of ^13^C-labeled malate by malic enzyme (see arrow in Figure 3) or conversion of ^13^C-labeled oxaloacetate by other enzymes [10,11]. The labeling % for lactate was almost negligible.

The large drop in labeling efficacy between fumarate/malate and citrate/isocitrate may be explained by (a) the distance in the biochemical pathway to the labeling source ^13^C-succinate, (b) dilution of labeled malate (precursor for subsequent TCA intermediates) in the interstitial space and diffusion away from the probe, and (c) that trafficking of energy substrates through the intercellular space is very limited—estimated to be less than 12% for glucose and lactate (see [12]) for a discussion of this topic).

The rapid decline in labeling efficacy using retrograde dialysis in vivo is unlike in vitro experiments where ^13^C-labeled precursors are added to the culture medium, yielding much higher labeling efficacy in subsequent TCA metabolites [13].

### 3.2. The Malonate Model

Malonic acid (malonate) is a reversible inhibitor of SDH, the enzyme converting succinate to fumarate. Another microdialysis study in rat brain using flow injection analysis with biosensors and perfusion with malonate (5–50 mM for 1 h) through the probe, reported rapidly increasing lactate and decreasing glucose levels in the dialysates [14], which is in line with mitochondrial inhibition and increased glycolysis. In the present study using similar doses of malonate (perfusion with 10 and 50 mM), however, we observed dramatic reductions in ^13^C-labeled and endogenous lactate and pyruvate, suggesting a strong inhibition of glycolysis. This discrepancy may be due to the different microdialysis membranes (15 kD cut-off PES membrane in the previous study versus 50 kD cut-off polyacrylonitrile (PAN) membrane in this study, which resulted in different recoveries and thus different interstitial malonate concentrations), and the different rat strains (Wistar versus Sprague Dawley) used in these studies. It has been reported that lower doses of malonate (i.e., 30 mM) inhibit the TCA cycle with only a partial effect on glycolysis whereas higher doses of malonate (i.e., 60 mM) inhibit both glycolysis and TCA cycle in rat skeletal muscle [15].

In our study, the effect of malonate perfusion is clearly illustrated by the dramatic rise in endogenous succinate levels and return to baseline levels when perfusion is switched to normal Ringer’s solution. The dose-dependent rise in endogenous succinate is perfectly in line with inhibition of SDH. The inhibition of *de novo* synthesis of fumarate and malate is most clearly illustrated by the complete inhibition of ^13^C incorporation in fumarate and malate during malonate perfusion, whereas reductions in endogenous levels are more modest with 10 mM malonate. The finding that endogenous malate levels are more affected by malonate perfusion than fumarate, although fumarate is the next intermediate in the TCA cycle after succinate, suggests that some back-cycling occurs between malate and fumarate [13], or that fumarate is produced from other sources, e.g., via the urea cycle [16]. Differences in the percentage change of ^13^C-labeled and endogenous TCA cycle intermediate were also apparent for citrate/isocitrate and illustrate that monitoring of ^13^C-labeled metabolites during perfusion with ^13^C-succinate is a much more sensitive tool to detect mitochondrial dysfunction than monitoring endogenous metabolite levels. Levels of ^13^C- and endogenous citrate during recovery after the first period of SDH inhibition also showed differences: A rebound effect (above baseline) for endogenous citrate, but still reduced levels for ^13^C-labeled citrate, suggesting increased activity of pyruvate dehydrogenase and pyruvate carboxylation to enhance levels of non-labeled oxaloacetate. Enhanced pyruvate carboxylation in neural tissue has been reported following irreversible inhibition of SDH using 3-nitropropionic acid [17]. Under normal conditions, pyruvate carboxylation only occurs in astrocytes, which has been studied previously using ^13^C-labeled bicarbonate [18].

Glutamine, which is related to the TCA cycle via glutamate (not detected) and alpha-ketoglutarate is released by astrocytes and taken up by neurons for glutamate synthesis and energy metabolism [19]. In the present study, ^13^C-incorporation in glutamine was completely blocked by malonate perfusion, reflecting strong inhibition of the TCA cycle in astrocytes (no uptake and no production of the precursor ^13^C -glutamate), as well as in neurons (no release of ^13^C -glutamate).

### 3.3. The ET-1-induced Transient Ischemia Model

(ET-1) is a potent vasoconstrictor, which has been associated with cerebral vasospasms and subsequent transient ischemic events in subarachnoid hemorrhage patients [20]. In rodents, intracerebral application of ET-1 in the vicinity of the medial cerebral artery has been used as an animal model for transient focal cerebral ischemia [21,22,23]. In this model, transient occlusion of the medial cerebral artery can be induced in awake, freely moving animals, causing ischemia-reperfusion injury in the ipsilateral striatum.

Recently, we described mitochondrial dysfunction in the ET-1 rat model, which was characterized by a prolonged elevation of the L/P ratio and concomitant normal or elevated levels of pyruvate following ischemia-reperfusion using an enzymatic assay with sampling time intervals of 15 min [23]. Here, using sampling intervals of 30 min and LC-MS, we observed a more than 50% drop of glucose in the first dialysate fraction following ET-1 application (at 30 min). In addition, we observed increases in glucose-6 phosphate (maximum increase about 7.5-fold at 30 min reperfusion, i.e., 60 min after ET-1 infusion) and lactate (maximum increase about 2.5-fold at 30 min after ET-1 infusion) lasting for up to 5 h after the insult, indicating degradation of brain glycogen and downstream glycolysis during and after the ischemic insult.

The ischemic insult caused a dramatic rise in endogenous succinate—up to a 5-fold increase at 30–60 min after onset of the insult. Ischemic succinate accumulation arises from reversal of SDH activity, which is driven by fumarate overflow from purine nucleotide breakdown and partial reversal of the malate/aspartate shuttle [24]. After reperfusion, re-oxidation of succinate by SDH may drive extensive reactive oxygen species production because of reverse electron transport at complex I. It has been reported that decreasing succinate accumulation by an SDH inhibitor, such as malonate (see above), can reduce ischemia-reperfusion injury in mouse models of heart attack and stroke [24,25]. In this context, it is interesting that cerebral perfusion with high doses of succinate has been proposed as a treatment to improve outcome in head injury patients [7].

We saw a large increase in endogenous malate levels peaking at 60 min after ET-1, concomitant with a significant drop in ^13^C-labeled malate at 30 min after ET-1, followed by elevated ^13^C-malate levels following reperfusion. Changes in ^13^C- and ^12^C-fumarate showed a similar pattern. The clinical significance of these changes may be as follows: If extracellular levels of ^13^C labeled metabolites are decreased and endogenous metabolites are increased, this may indicate compromised TCA cycle function (because of reduced labeling %) and damage to mitochondrial and cellular membranes (because of the increase in endogenous metabolite levels).

Most pyruvate is formed by glycolysis, but a minor part can be formed by conversion of malate by malic enzyme (ME) activity. In this study, ^13^C-labeled pyruvate is thus derived from ^13^C-labeled malate. In contrast to endogenous pyruvate levels, which were not significantly changed by the ET-1 insult, ^13^C-labeled pyruvate levels were increased up to a maximum of 2.5-fold of baseline at 30 min of reperfusion (i.e., 60 min after ET-1 infusion) and were still elevated above baseline levels for the next 4 h (Figure 3). However, these changes, expressed as % labeling did not reach significance (Table 1). Increased ME activity shortly after the insult may play a role in combating oxidative stress [26].

In contrast to fumarate and malate, endogeneous citrate/isocitrate levels decreased in the first 30 min after onset of ischemia, which is in line with reduced influx of acetyl-CoA into the TCA cycle during the period of compromised cerebral blood flow. However, immediately after the period of ischemia, endogeneous citrate/isocitrate levels started to rise to about 2-fold of baseline, whereas ^13^C-labeled citrate increased by up to 3–4 times, indicating a faster running TCA cycle after ischemia.

Opposite to the malonate model, where dramatic changes in alpha-ketoglutarate are paralleled by changes in glutamine (Figure 2), in the ET-1-induced ischemia-reperfusion model glutamine levels did not follow the elevations in alpha-ketoglutarate levels (Figure 3). Glutamate-glutamine cycling, i.e., glial uptake of glutamate, conversion to glutamine and subsequent release of glutamine, is an energy demanding process, which is known to be impaired following ischemic events. A microdialysis study in neurointensive care patients with subarachnoid hemorrhage reported an inverse correlation between low glutamine/glutamate ratios and elevated L/P ratios [27]. Thus, the lack of significant changes in endogenous or ^13^C-labeled glutamine while alpha-ketoglutarate is transiently increased after the insult, may be explained by impaired energy metabolism in the reperfusion phase.

## 4. Conclusions

Following perfusion with ^13^C-succinate, changes in ^13^C-labeled TCA metabolites provide a more sensitive index of TCA cycle dysfunction, than changes in endogenous TCA metabolites. Discrepancies between extracellular changes in ^13^C-labeled and endogenous metabolites under pathological conditions may be explained by the loss of cell membrane integrity (cell death). The differential response of endogenous versus ^13^C-labeled malate can be used to monitor metabolic perturbations following cerebral ischemia-reperfusion.

Microdialysis-perfusion with a tracer dose of ^13^C-succinate and subsequent LC-MS analysis of dialysate fractions is a promising research tool to monitor neurointensive care patients and get in-depth information on TCA cycle dysfunction following vascular or traumatic brain insults.

## 5. Materials and Methods

### 5.1. Animals

The animal experiments were approved by the local ethics committee and in accordance with the Danish Animal Experiment Inspectorate and EU legislation (lic. Nr. 2017-15-0201-01256). A total of 18 adult Sprague Dawley rats were used, weighing on average 273 g (range 216–379 g) with a mean age of 7.5 weeks (range 6–9 weeks) on the day of stereotaxic surgery (see 5.2 below). After surgery the rats were individually housed in a 12 h light/dark cycle with free access to food and water.

Eight rats (purchased from Janvier labs, Saint-Berthevin, France) were used for the malonate perfusion experiments (see 5.4 below). Two malonate-treated rats were used for optimization of LC-MS analysis of dialysates (see 5.7 below) and the data for one other rat were useless because of technical problems with the LC-MS, leaving 5 malonate-treated rats for statistical analysis.

Ten rats (purchased from Taconic Biosciences A/S, Ejby, Denmark) were used for ET-1 experiments (see 5.2 and 5.3 below). Four ET-1 treated rats were discarded, because glucose levels were not reduced following ET-1 administration, indicating that induction of cerebral ischemia was unsuccessful. Two other ET-1 rats were discarded from the statistical analysis because of flow problems during microdialysis. Thus, four ET-1 rats were included in the statistical analysis of the data.

### 5.2. Stereotaxic Surgery

For the ET-1 experiments, two microdialysis guide cannulas (shaft: 4 mm, Brainlink^®^, Groningen, Netherlands) were implanted in the left cerebral hemisphere using a stereotaxic frame (Kopf Instruments, Tujunga, CA, USA). One guide was placed for microinjection of ET-1 into the piriform cortex close to the proximal part of the medial cerebral artery (MCA) and one guide was placed for microdialysis in the ipsilateral striatum (see Figure 4B). For experiments with the SDH inhibitor malonate only one microdialysis guide was placed in striatum. In these experiments, malonate was administered by perfusion through the microdialysis probe (retrograde dialysis). The stereotaxic coordinates relative to bregma, with the nose bar at −3.9 mm (according to the atlas of Paxinos and Watson, 1986), were as follows:

Guide cannula for ET-1 injection: A + 0.9 mm; L 5.2 mm; V 4.6 mm (ET-1 experiment only)

Guide cannula for microdialysis probe: A + 0.5 mm; L 2.5 mm; V 3.2 mm

Stereotaxic surgery was done under Hypnorm/Dormicum anesthesia (Hypnorm: 0.315 mg/mL fentanyl and 10 mg/mL fluanisone, Janssen Pharmaceutica, Beerse, Belgium; 0.3 mL/kg s.c. Dormicum: 5 mg/mL midazolam, Hoffmann-La Roche, Basel, Switzerland; 5 mg/kg s.c.). Lidocaine (20 mg/mL Farmaplus AS, Oslo, Norway) was used as a local anesthetic. Body temperature was kept at 37.5 °C with a thermostatically regulated heating pad (Bosch CTKI3, München, Germany).

The guide cannulas and a slotted screw for head block tethering (Instech labs Inc., Plymouth Meeting, PA, USA) were fixed to the skull using glass ionomer luting cement (GC Fuji plus capsule, GC corporation, Tokyo, Japan). A slow release oral formulation of 0.4 mg/kg buprenorphin (Temgesic 0,2 mg sublingual tab., RB Pharmaceuticals, Slough, UK) was used as postoperative analgesia and rehydration was administered as a subcutaneous injection of 5 mL 0.9% NaCl immediately after surgery.

### 5.3. Microdialysis Setup for ET-1-Experiments

One day after stereotaxic surgery, the microdialysis probe (50kDa cut-off, 3mm polyacrylonitrile (PAN) membrane, BrainLink^®^, Groningen, The Netherlands) and probe for ET-1 injection (see above) were inserted through the guide cannulas under brief anesthesia (ca. 5 min) using inhalation of isoflurane (Baxter A/S, Allerød, Denmark). The inlet and outlet tubing (FEP tubing, 1.2 mL/10 cm, AgnTho’s AB, Stockholm, Sweden) was connected to a swivel (AgnTho’s AB, Stockholm, Sweden), syringe pump (22 Harvard Apparatus, Inc., Holliston, MA, USA) and fraction collector (CMA 142, Stockholm, Sweden) using 0.38 mm IDEX silicon connectors.

Within approximately 15 min after insertion through the guide, the microdialysis probe was perfused with 1 mM ^13^C-labeled succinate (^13^C_4_ 99%, Sigma-Aldrich, Denmark A/S, Copenhagen) in sterile Ringer’s solution at a flow rate of 1.0 µL/min. Microdialysis fractions were collected at 30 min intervals and stored at −20 °C within 2 h after collection. Microdialysis experiments were performed in awake, freely moving animals.

The ET-1 injection cannula (an old microdialysis probe of which the membrane was removed) was connected to a 100 µL Hamilton syringe using FEP tubing. The Hamilton syringe and FEP-tubing were filled with an ET-1 solution (Endothelin-1 ≥ 97%, Sigma-Aldrich Denmark A/S, Copenhagen, 10 pmol/µL dissolved in sterile Ringer’s solution (147 mM NaCl, 4 mM KCl, 1.1 mM CaCl_2_, 1.0 mM MgCl_2_) for manual infusion of 15 µL ET-1 solution. After insertion of the ET-1 injection cannula, the tip of the fused silica tubing ended 3.0 mm below the guide cannula, i.e., 7.6 mm ventral to bregma. ET-1 was infused after collection of the first six 30 min fractions (baseline monitoring).

### 5.4. Mitochondrial Inhibition by Malonate Perfusion

Malonate perfusion experiments were done one day after stereotaxic placement of a guide cannula in striatum (unilaterally) and after inserting the microdialysis probe through the guide cannula as described above. In the malonate experiments, 60 min fractions (flow rate 1.0 µL/min) were collected and baseline levels were monitored for six hours using Ringer’s solution with ^13^C-labeled succinate (6 samples of 60 µL) before starting perfusion with 10 mM malonate (malonic acid, disodium salt monohydrate, Sigma-Aldrich, dissolved in Ringer’s containing ^13^C-succinate) for another six hours, followed by regular Ringer’s with 1 mM ^13^C-succinate for eight hours and finally six hours perfusion with 50 mM malonate in Ringer’s containing 1 mM ^13^C-succinate.

### 5.5. ET-1 Induced Transient Cerebral Ischemia

After three hours of baseline monitoring collecting 30 min fractions (6 samples of 30 µL), focal transient ischemia was induced by infusing 150 pmol ET-1 (Sigma-Aldrich) in 15 min (10 pmol/µL, 1 µL/min; ET-1 dissolved in sterile Ringer’s solution; 60 µL aliquots of 10 pmol/µL were stored at −20 °C). Microdialysis was continued for at least six hours after the insult.

#### Histology

One day after microdialysis, the rats were killed by a lethal dose of pentobarbital (pentobarbital 200 mg/mL with lidocainehydrochloride 20 mg/mL, Glostrup Apotek, Denmark, 0.2–0.3 mL pr. rat) and decapitation before cardiac arrest. The brains were rapidly removed from the skull, and frozen using high pressure CO_2_ and stored at −80 °C until histological processing (cryostat sectioning and toluidine blue staining) for analysis of the placement of guides and infarct size.

### 5.6. Statistical Analysis

Metabolite data are expressed as mean +/− SEM of percentage change of each rat’s own baseline and visualized as a time-line. Statistical differences between groups were analyzed using nonparametric Kruskal Wallis test with Dunn’s multiple comparison’s test (vs. baseline). The XY graphs were generated in GraphPad Prism (GraphPad Software, inc. San Diego, CA, USA).

### 5.7. LC-MS Sample Preparation and Analysis

Each microdialysis fraction of 30 µL was lyophilized prior to resuspension in 12 µL 1% formic acid (FA) and transfer to vial including 100 µL insert. A pool (quality control) of the samples was constructed by transferring 2 µL of each sample to a new vial, which was injected in every sixth sample to monitor signal drift and system reproducibility. Ten µL from each sample was injected in random order using a 1290 Infinity high pressure liquid chromatography system (Agilent Technologies) equipped with a Supelco Discovery^®^ HS F5-3 (2.1 × 150 mm and 3 µm particle size) column kept at 40 °C. Compounds were eluted using a flow rate of 300 µL/min and the following gradient composition of A (0.1% FA) and B (0.1% FA, acetonitrile) solvents: 100% A from 0–3 min, 100–60% A from 3–10 min, 60–0% A from 10–11 and 100% B isocratic from 11–12 min before equilibration for 5 min with the initial conditions. Eluting metabolites were detected by a 6530B quadrupole time of flight mass spectrometer (Agilent Technologies) operated in negative ion mode scanning from 40–1050 m/z with the following settings: 3 scans/sec., gas temp at 325 °C, drying gas at 8 L/min, nebulizer at 35 psi, sheath gas temp at 350 °C, sheath gas flow at 11 L/min, VCap at 3500 V, fragmentor at 125 V and skimmer at 65 V. Each spectrum was internally calibrated during analysis using the signals of purine (119.03632) and Hexakis 1H,1H,3H-tetrafluoropropoxy phosphazine with formate adduct (966.000725), which was delivered to a second needle in the ion source by an isocratic pump running with a flow of 20 µL/min. A library containing molecular formula and retention time of the metabolites of interest was constructed using MassHunter PCDL Manager v. B.08.00 (Agilent Technologies). All reported annotations, except lactate and glutamine, were based on accurate mass and co-elution with synthetic standards and their fragments (Metabolomics Standards Initiative (MSI) [28] level 1 annotation). Lactate and glutamine were annotated based on the existence of co-eluting fragments from a pooled sample analyzed in “all-ion” mode using 0, 10 and 40 V in collision energy (MSI level 3 annotation). The ion fragments from the known compounds were obtained from METLIN [29]. Chromatograms for all compounds were extracted and the areas were quantified using Profinder v. B.08.00 (Agilent Technologies) in “Batch isotopologue extraction” mode, which extracts the signal from the isotopes and corrects for their natural abundance, with a mass tolerance of 10 ppm and retention time tolerance of 0.1 min. Quality control samples were used to evaluate system reproducibility, and potentially, to exclude compounds with a relative standard deviation (RSD) above 30%, however, all shown compounds had a RSD < 15%.

## Figures and Tables

**Figure 1 metabolites-09-00204-f001:**
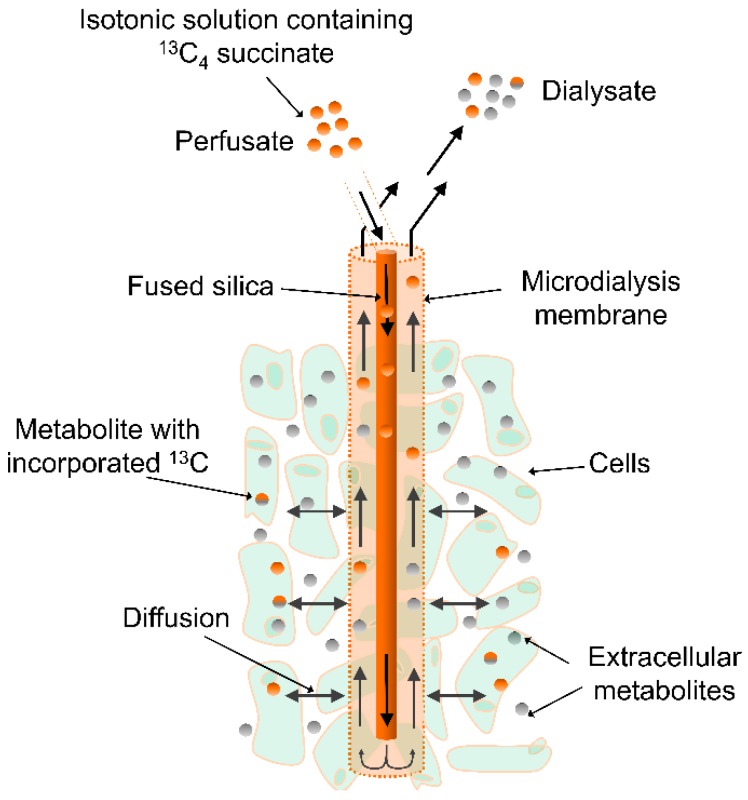
Microdialysis set-up. Ringer’s solution with labeled ^13^C-succinate is delivered to the microdialysis probe in the rat brain. Diffusion through the membrane allows ^13^C-succinate to be taken up and metabolized by the surrounding cells.

**Figure 2 metabolites-09-00204-f002:**
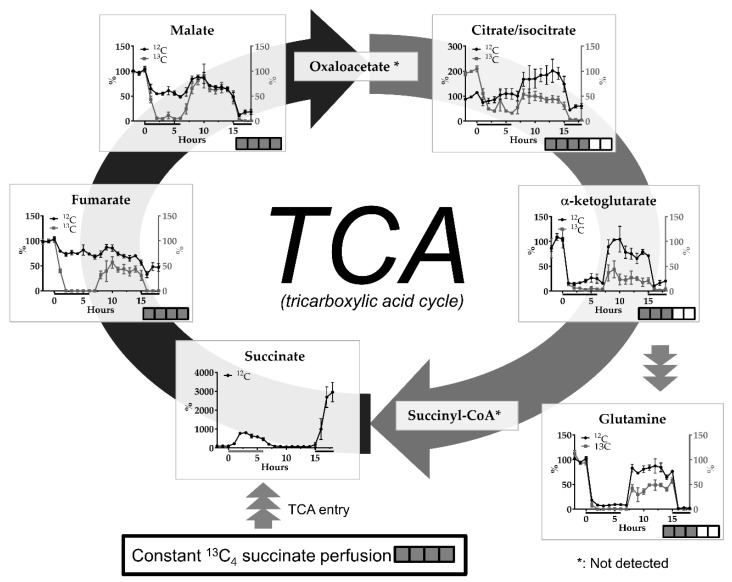
Effects of perfusion with 10 mM (0–6 h) or 50 mM malonate (14–18 h) on interstitial levels of endogenous (^12^C) and ^13^C-labeled tricarboxylic acid cycle (TCA) metabolites succinate, fumarate, malate, isocitrate, alpha-ketoglutarate and related glutamine upon constant perfusion with 1 mM ^13^C_4_ succinate. Data are mean +/− SEM of 5 rats and expressed as % change of baseline abundance. The number of ^13^C or ^12^C atoms in the different metabolites after uptake of ^13^C_4_ succinate and one turn of the TCA cycle is indicated with the number of filled or open squares. Black bars under the x-axis illustrate time periods of malonate administration. * not detected.

**Figure 3 metabolites-09-00204-f003:**
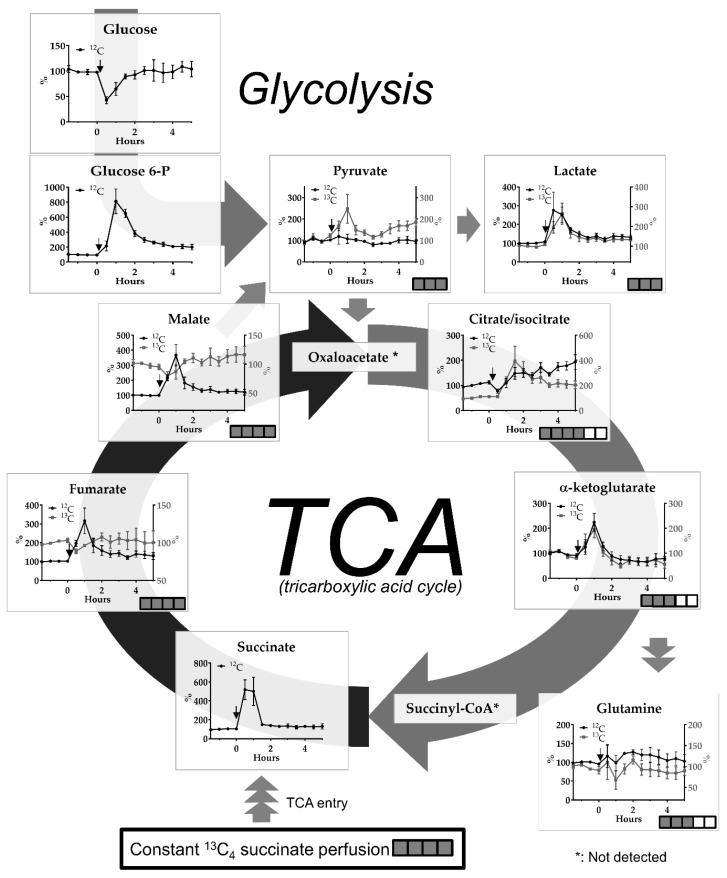
Effect of ET-1-induced cerebral ischemia-reperfusion on interstitial levels of endogenous (^12^C) glucose and glucose-6-phosphate and endogenous (^12^C) and ^13^C-labeled pyruvate, lactate and TCA metabolites succinate, fumarate, malate, isocitrate, alpha-ketoglutarate and related glutamine upon constant perfusion with 1 mM ^13^C_4_ succinate. Data are mean +/− SEM of 4 rats and expressed as % change of baseline abundance. Number of ^13^C or ^12^C atoms in the different metabolites is indicated with number of filled or open squares. Black arrows indicate time point of ET-1 infusion. * means not detected.

**Figure 4 metabolites-09-00204-f004:**
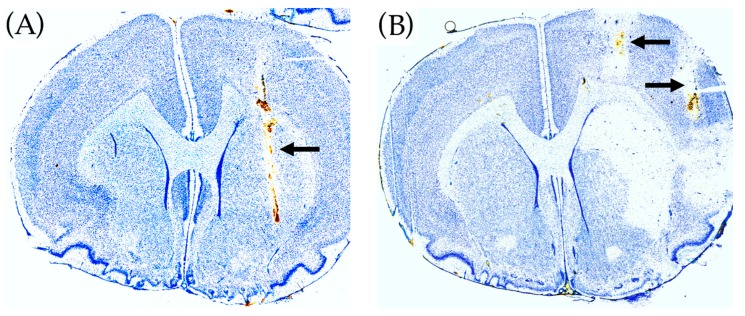
Histology of rat brains using toluidine blue staining. (**A**) Malonate perfusion model. The position of the microdialysis probe in striatum is shown by the arrow. (**B**) ET-1 rat model of transient cerebral ischemia. The position of the guide cannula for the microdialysis probe in the ipsilateral striatum is shown by the upper arrow (the microdialysis probe track in striatum is not visible). The position of the guide cannula for ET-1 infusion in the pirifom cortex is shown by the lower arrow. ET-1 infusion caused histological damage in the ipsilateral striatum.

**Table 1 metabolites-09-00204-t001:** ^13^C-labeling % for selected metabolites in the ET-1 model. Data is shown as mean +/− SEM of 4 rats. Significant differences using nonparametric Kruskal Wallis test with Dunn’s multiple comparison’s test (vs. baseline) are shown: ** *p* < 0.005 and **** *p* < 0.0001.

% Labeling	Fumaric Acid	Malic Acid	(Iso) Citrate	α-Ketogluarate	Glutamine	Pyruvate	Lactic Acid
Baseline	88.1 (+/− 0.2)	91.3 (+/− 0.4)	8.4 (+/− 0.9)	14.0 (+/− 2.0)	6.1 (+/− 0.6)	6.3 (+/− 1.3)	0.3 (+/− 0.04)
Ischemia + 30 min of Reperfusion	79.2 (+/− 2.1) **	75.3 (+/− 2.6) ****	11.9 (+/− 2.8)	15.9 (+/− 2.7)	4.7 (+/− 0.7)	8.7 (+/− 1.4)	0.2 (+/− 0.02)
After >4 h Reperfusion	87.5 (+/− 0.5)	91.1 (0.4)	9.6 (+/− 1.0)	14.1 (+/− 3.0)	5.3 (+/− 0.8)	8.4 (+/− 1.3)	0.3 (+/− 0.03)

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
