# Peer review of "In Vivo Microdialysis of Endogenous and ^13^C-labeled TCA Metabolites in Rat Brain: Reversible and Persistent Effects of Mitochondrial Inhibition and Transient Cerebral Ischemia"

_metabolites, 2019, doi:10.3390/metabo9100204_

Round 1

Reviewer 1 Report

The authors present an innovative and well-written manuscript. Using cerebral microdialysis, they established a new model to investigate key-metabolites of the tricarboxylic acid (TCA) cycle during induced mitochondrial dysfunction and transient cerebral ischemia in the awake animal. Continuous microdialysis catheter perfusion with 13C-labeld succinate during their experiments allows to differentiate between de-novo synthesis and endogenous release of metabolites. This topic is of particular interest as it provides improved insight into pathophysiological processes of TCA cycle dysfunction, which are often encountered in the clinical setting.

Reviewing your manuscript, the following questions have emerged:

Table 1: Are the results shown as the ratio 13C/12C or “%”.? In the result section it is defined as ratio (12C/13C) and in the table as “% labeling of 13C/12C “(reciprocal ratio)!? What does e.g. 748% mean? Percentage of what? Or is it the ratio? Please clarify. Why did you choose a t-test and not a nonparametric test, as a sample size of 4 seems not to be normally distributed? Please indicate your statistic tests as well in the “Statistical analysis” section. Figure 2&3: What does the asterix “*” next to “Oxaloacetate” and “Succinyl-CoA” mean. Please indicate in the Figure caption. An arrow at the time when malonate was administered (hour 0 and 5) or a bar for the time period of application would improve illustration in each graph. Histological brain damage: Was there a difference of the ET-1 induced ischemic damage between the animals? Did you observe any damage/bleeding/inflammation next to the microdialysis probe, which might have influenced your results? The sentence “Microdialysis LC-MS data of the malonate model and ET-1 model are shown in Figure 2 and Figure 3, respectively.” should be deleted as these figures are not the topic of this section. The discussion is well written and provides a point-by-point explanation of the results of both models. However, the most interesting part of your experiment are the specific metabolic differences between both models, especially regarding the differences of endogenous versus 13C-labeled metabolites? Elaborate/summarize more (in a separate section) the discrepancies of the ET-1 and malonate-model as outlined in your abstract. Are there any clinical implications? As outlined in the introduction, the L/P-ratio is used in the clinical setting as the most sensitive ischemic parameter, which is less specific as it may rise as well during episodes of mitochondrial dysfunction. Does your experiment provide any new information on that? Are there any new possibilities for better differentiation of these completely divergent metabolic states? Methods: Please indicate the number of animals used during your experiments in the “Animals” section and not in the “Statistical analysis” section. Please include the total number and number of included animals for your experiment. Indicate reasons for dop outs. In Line you refer to “fig. x”. Do you mean Figure 3? Microdialysis setup for ET-experiements: The guide cannula was inserted one day before the experiment. How long was the interval between insertion of the microdialysis probe and start of sampling for the experiment? Why did you choose different baseline intervals (six hours for malonate and 3 hours for ET-1)? Statistical analysis: Insert “.” after time-line (Line 340) Supplementary Materials: Remove this paragraph if you do not provide any of the mentioned materials. Abbreviations: Please follow the policy of the journal and define all abbreviations the first time when they appear in the abstract, main text, and figures: TCA is defined in Line 53, but appears first in Line 48 Please define: NMR (Line 55) LC-MS (Line 64) ROS (Line 224) ME (Line 232) NADP (Line 233) HPLC (Line 344) Endothelin-1 is abbreviated in the Discussion and Methods section, but not in the Result section; sometime it is abbreviated as “Et-1” and sometimes as “ET-1”. The abbreviation is explained at multiple Lines throughout the manuscript. Only define your abbreviations at the first time they appear. Define the abbreviation (TCA) in Figure 2&3

Reviewer 2 Report

General comments:

This paper describes an interesting application of microdialysis and retrodialysis in analysis of metabolites and mitochondrial functionality. It combines metabolic tracing using infusion of stable isotope labeled metabolites with targeted LC-MS analysis of selected metabolites of both endogenous as well as labeled metabolites.

Data are well presented graphically and generally seems to make sense from a biological point of view – though we are not experts within deciphering the complex data obtained from microdialysing brains. Accordingly, our review mainly will focus on the methodological parts of the publication. Since the authors claim to describe a novel approach to monitor mitochondrial dysfunction we believe the paper lacks a more thorough description of method development and validation data. Regarding the the discussion, we generally find it somewhat disorganized (particular for the second ET-1 model and occasionally also unclear. It will benefit from shortening/ aggregating of the text and careful use of right terms e.g. do the authors mean co-enzyme A or acetyl CoA etc.

Materials and methods:

In general, a good description of the technical parts (material and methods).

However, there is no description of validation of the LC-MS method (e.g. estimates of LOD/LOQ, precision and trueness, linearity etc.). Did the authors perform any validation of the analytical LC-MS method? It is no-where mentioned what levels of the analytes are actually measured nor put in relation to the capabilities of the method.

How reliable and reproducible is quantification just using peak area from extracted ion chromatograms?

The authors should present some evaluation of the performance of their LC-MS method.

Optimization of microdialysis sampling methodology (flow and recovery rates) should be described and commented. It is discussed that differences in microdialysis probes makes a potentially great impact on the results in malonate model.

Results:

The labeling efficiency data in table 1: Authors are adviced to comment more in detail about the gross differences in isotope labeling efficiency of malate+fumarate versus downstream TCA metabolites, is this reasonable data?

All data are presented in a relative format. Presentation of absolute quantitation data (i.e. concentrations) would be interesting and a significant improvement. Such data is obtainable when using a calibrated LC-MS method and microdialysis recovery rate correction (even though we acknowledge that they still are only estimates of the true interstitial concentrations).

Lactate/pyruvate (L/P) ratio mentioned often in text as a marker of mitochondrial dysfunction but (L/P) ratios were not presented in detail in this study, why not?

Figure 2 and 3: Markings on the x-axis indicating the timing of the different experimental events e.g. start/stop malonate perfusion as well as timing of ischemia/reperfusion phases would make plots easier to decipher. The use of the quite small font in the drawings also makes interpretation of labels difficult e.g. carbon 12 eller 13.

Figure 3: Describes the metabolite response following a ischemia reperfusion event. The observed metabolite changes and C12/C13 differences do not convey a clear message, and the accompanying discussion of these data is not completely convincing.

Discussion:

Disproportionally long section. Would benefit from shortening and focusing.

Round 2

Reviewer 1 Report

After revision no further comments.

Author Response

Thank you for the review

Reviewer 2 Report

Response to revised version:

General comments:

The authors have presented a revised manuscript with several improvements e.g. analytical method reproducibility (RSD) using quality control samples and microdialysis flow rate are now described as well as graphics in the result section are now clearer.

Furthermore, they have commented on many of the question raised in our initial evaluation.

Regarding our major point concerning analytical method validation, the authors argue that a full validation of the analytical method (untargeted LC-MS) is beyond the scope of their study. We agree that this may be beyond the scope of the present study, but on the other hand also note that no evidence is presented that the authors indeed measure what they claim to measure. At least it is custom even in metabolomics studies to validate that the metabolites reported are indeed the correct metabolites and not some other potential isomers. So our advice is that they at least provide some documentation that they indeed measure the correct metabolites. This can be done by analyzing authentic standards and comparing retention times and fragments to the tentative metabolites measured in the paper. If they do not have the isotopically labeled standards, then they may use non-labeled references as substitute (though less optimal).

We will leave it to the editor to evaluate the discussion.
